# Complications of Reverse Total Shoulder Arthroplasty: A Computational Modelling Perspective

**DOI:** 10.3390/jcm10225336

**Published:** 2021-11-16

**Authors:** Yichen Huang, Lukas Ernstbrunner, Dale L. Robinson, Peter Vee Sin Lee, David C. Ackland

**Affiliations:** 1Department of Biomedical Engineering, University of Melbourne, Parkville, VIC 3010, Australia; yichenh2@student.unimelb.edu.au (Y.H.); lukas.ernstbrunner@gmx.at (L.E.); d.robinson@unimelb.edu.au (D.L.R.); pvlee@unimelb.edu.au (P.V.S.L.); 2Department of Orthopaedic Surgery, Royal Melbourne Hospital, Parkville, VIC 3050, Australia; 3Melbourne Orthopaedic Group, Windsor, VIC 3181, Australia

**Keywords:** reverse shoulder prosthesis, finite element model, biomechanical model, scapular notching, joint instability, loosening, acromion fracture

## Abstract

Reverse total shoulder arthroplasty (RTSA) is an established treatment for elderly patients with irreparable rotator cuff tears, complex proximal humerus fractures, and revision arthroplasty; however, with the increasing indications for RTSA over the last decade and younger implant recipients, post-operative complications have become more frequent, which has driven advances in computational modeling and simulation of reverse shoulder biomechanics. The objective of this study was to provide a review of previously published studies that employed computational modeling to investigate complications associated with RTSA. Models and applications were reviewed and categorized into four possible complications that included scapular notching, component loosening, glenohumeral joint instability, and acromial and scapular spine fracture, all of which remain a common cause of significant functional impairment and revision surgery. The computational shoulder modeling studies reviewed were primarily used to investigate the effects of implant design, intraoperative component placement, and surgical technique on postoperative shoulder biomechanics after RTSA, with the findings ultimately used to elucidate and mitigate complications. The most significant challenge associated with the development of computational models is in the encapsulation of patient-specific anatomy and surgical planning. The findings of this review provide a basis for future direction in computational modeling of the reverse shoulder.

## 1. Introduction

Reverse total shoulder arthroplasty (RTSA) has traditionally been used in the treatment of patients with a rotator cuff-deficient arthritic shoulder. However, over recent years, indications for RTSA have expanded to include irreparable rotator cuff tears, complex proximal humerus fractures in elderly patients, severe glenoid bone loss, revision arthroplasty, and tumour resection. In the United States, the number of primary RTSAs has almost tripled from 2012 to 2017, accounting for around 60% of all total shoulder arthroplasty procedures in 2017 [1]. In Australia, over 80% of total shoulder arthroplasty procedures performed in 2019 were RTSA [2].

RTSA, originally designed by Professor Paul Grammont in 1985, reverses the ball-and-socket anatomy of the anatomical glenohumeral joint [3]. This has the effect of medialising the centre of rotation of the glenohumeral joint, thereby increasing the moment arms (leverage) of the deltoid and facilitating the recruitment of more of its fibres during elevation [4,5,6]. The semi-constrained joint articulation confers stability, compensating for absence of rotator cuff muscles to a degree, while the prosthesis design also distalizes the humerus which tensions the deltoid, ultimately allowing for increased force generation and range of joint motion [7].

Despite the biomechanical benefits and widespread use of RTSA, complication rates remain a concern [8,9,10,11,12,13,14,15,16]. A recent study of 4124 shoulders with RTSA reported a complication rate of 16.1% [17]. Shoulder instability, periprosthetic fracture, infection, and component loosening are the leading causes of revision RTSA [17,18]. Scapular notching, acromial fractures, instability, and component loosening are a consequence of post-operative joint biomechanics and can be directly attributed to implant design and placement. The need for a deeper understanding of muscle and joint function after RTSA, to inform strategies to mitigate complication, has motivated the development of more sophisticated computational models of RTSA. The objective of this paper is to review computational modeling strategies for evaluation of complications associated with RTSA. Firstly, computational modeling techniques used in simulating RTSA were identified. Secondly, models used in evaluating complications of RTSA were discussed with respect to these modeling and simulation techniques. Third, validation of computational models and benchmarking of model-generated data with clinical results were discussed. Finally, an overview of the major limitations of current computational models of RTSA and directions of future computational modeling were presented.

## 2. Computational Modeling Techniques

Computational modeling of RTSA facilitates estimation of muscle and joint loading, which is currently impossible to measure non-invasively in vivo. Modeling and simulation of RTSA has played a critical role in guiding implant positioning and surgical technique [19,20,21,22], prosthesis selection [23,24], implant design [25,26], and post-operative rehabilitation prescription [27,28]. To date, most modeling and simulation tools require high levels of expertise and are generally restricted to the research setting.

Biomechanical computer models are broadly categorized into three groups: rigid body models, finite element (FE) models, and multi-body models. Rigid body models characterize bones as non-deformable segments. Through simulating joint kinematics and internal and external forces, these models are used to investigate changes in muscle and joint function after RTSA [29,30]. The primary advantage of rigid body models is that they can run quickly and at a low computational cost. For example, shoulder muscle forces during upper limb elevation can be calculated in several minutes using a subject-specific rigid body model (Figure 1A) [28,31]; however, since deformation mechanics is neglected, the internal stresses and strains in the bone and implant cannot be accurately evaluated using this method. An important utility of rigid body models has been in their capacity to estimate the influence of arthroplasty on muscle moment arms, muscle and joint forces, and range of motion (ROM) at the shoulder, which are primary indicators of implant functional performance [21,23,24,29,30,31,32,33,34,35,36,37,38,39,40].

In contrast, FE models provide estimates of material deformation by discretisation of structures into a finite number of elements, each element of which can be interrogated for load response such as stress, strain and displacement. This may reveal, for instance, how joint load is transmitted to bone via joint contact loading. The FE method has been used to investigate wear, component fixation, and joint stability after RTSA [41,42,43,44,45], which has aided in implant selection, intraoperative positioning, and fixation [20,44,46,47]. However, in contrast to rigid body models, FE model simulations can be computationally expensive. A FE model of the shoulder after RTSA, containing around half a million elements (Figure 1B), takes more than thirty minutes to run using a standard desktop computer [28].

Finally, multi-body musculoskeletal models input muscle and/or joint reaction force estimations from a secondary model simulation, typically derived from a rigid body model, into a FE analysis. This approach provides a basis for muscle and joint force prediction, as well as bone and implant deformation. Multi-body modeling has been used to assess the performances of different implant designs and surgical techniques in RTSA, with the capability to take into account accurate and physiologically meaningful boundary conditions [28,48,49,50,51].

## 3. Computational Modeling of Reverse Total Shoulder Arthroplasty Complications

### 3.1. Scapular Notching

Scapular notching refers to erosion of the scapular neck due to impingement with the humeral tray of the RTSA, and has a reported prevalence of between 20% and 96% [14,52,53]. Although the impact of scapular notching on clinical outcomes remains debated [54], it has been shown in some studies to be associated with lower constant scores, lower subjective shoulder values, lower active shoulder ROM, and revision surgery [14,52]. A recent study revealed that two types of impingements coexist after RTSA: (i) abutment type impingement, which restricts shoulder movement and may lead to fatigue fracture of the bone, and (ii) friction type impingement (scapular notching), associated with friction forces generated by contact between the polyethylene component and scapula, resulting in bone notching and subsequent erosion. Friction-type impingement has been found to be the result of implant movement in multiple directions, as opposed to just contact between the bone and the implant [55]. However, this phenomenon is virtually always simplified in computational modeling. Specifically, model simulation studies to date quantify scapular notching as a surface-to-surface bone contact or collision event during shoulder motion, primarily for the purposes of estimating impingement-free ROM (Figure 2).

One study evaluated the effect of glenosphere positioning on internal rotation and external rotation after RTSA [56]. Three-dimensional (3D) models of the scapula and humerus were reconstructed from computed tomography (CT) scans of seven cadaveric specimens, and the 3D models of the reverse prosthesis components used in simulated RTSA were obtained from laser scanning. The findings suggested that inferior translation, lateralisation, and inferior tilt of the glenosphere can increase internal rotation and external rotation range of motion, while superior translation and medialisation of the glenosphere constrains this degree of freedom. Another study used a similar modeling method to investigate the effects of various humeral stem designs on postoperative shoulder ROM and humeral position [26]. The results suggested that, when compared to the traditional Grammont prothesis, a 145° onlay stem improved overall shoulder ROM and only led to a limited change in humeral offset. This rigid body model was later used in a study to determine the effects of different glenoid configurations on humeral position, shoulder ROM, and rotator cuff muscle length [57]. It was found that an eccentric 36 mm glenosphere with a 145° onlay humeral stem produced the highest postoperative shoulder ROM of those investigated. A further study built rigid body models from CT scans of 20 patients with osteoarthritis to evaluate the impacts of humeral neck shaft angle and glenoid lateralization on ROM after RTSA [58]. The results showed that lowering humeral neck shaft angle and glenoid lateralization were both effective in increasing impingement-free ROM after RTSA. Another modeling study based on a dataset of 30 cadaveric specimens used similar modeling techniques to investigate how humeral version affects impingement after RTSA (Figure 2) [59]. It was found that humeral version greatly influenced scapular notching following RTSA. Specifically, humeral retroversion was found to increase ROM during abduction, flexion, scaption, and axial rotation, while neutral humeral version may improve ROM during activities of daily living. However, the large variability in the results across specimen models underscores the importance of subject-specific modeling in the investigation of scapular notching.

A more recent study used rigid body modeling to evaluate the combined effects of humeral stem design, humeral neck-shaft angle, glenoid lateralisation, and glenoid eccentricity on ROM after RTSA. It was found that only glenoid lateralisation significantly increases overall ROM in RTSA, and that the use of a semi-inlay 145° prosthesis with 4 mm lateralisation and 2 mm inferior eccentricity was the most suitable and universally applicable in preventing scapular notching after RTSA [60].

Prosthesis designs that aim to prevent scapula notching have also been investigated using FE modeling. A 2-dimensional (2D) FE model was developed to assess the effects of design variations of bony increased-offset reverse shoulder arthroplasty (BIO-RSA) on the glenoid component internal stresses (Figure 3) [61]. Cortical and cancellous bone was modeled, with the results suggesting that excessive eccentric and lateral offset of the glenosphere, though reducing scapular notching, may significantly increase the stress at the glenoid-baseplate interface. A later study used a 3D FE model to evaluate how glenoid-sided lateralisation, glenosphere size, baseplate orientation, and peripheral screw orientation influences the stress and displacement of the glenoid component [62]. It was found that stress and displacement increased with glenoid-sided lateralisation, and both can be reduced with a smaller glenosphere, inferior tilt of the baseplate, and divergent peripheral screws. Prosthetic lateralisation was also found to increase stress and displacement to a lesser degree than bony lateralisation.

### 3.2. Component Loosening

Component loosening is a complication of RTSA reported in between 1.8% and 4.8% of cases, with most cases associated with the glenoid component [17,63]. It was reported that component loosening is associated with up to 34% of revision surgeries [64]. The causes of this complication frequently relate to poor bone stock due to osteoporosis or osteolysis, component malpositioning, or scapular notching [17,65]. Computational models have been developed to investigate the influences of different prosthesis designs and intraoperative positioning on implant fixation after RTSA.

Some of the earliest studies of component fixation quantified micromotion at the bone-implant interface using FE modeling [66,67,68]. One FE study assessed the effects of lateralising the glenohumeral joint centre of rotation, screw insertion angle, screw length, screw diameter, and bone material quality on fixation of the glenoid component for three RTSA implant designs (Figure 4) [68]. The screws, baseplate, and glenosphere were fixed to the glenoid, which was modeled as a polyurethane block. Cyclic uniaxial loads were applied to the glenosphere to simulate joint loading, while bone-implant micromotion was evaluated. It was shown that, for all three designs, implant micromotion can be minimized by increasing the angle of screw insertion at the central axis of the implant, lengthening or thickening the screws, decreasing the lateral offset of the glenohumeral joint centre of rotation, and increasing the bone stiffness. An FE modeling study of similar design found that glenosphere undersurface contact with the glenoid can reduce both average stress and bone-implant micromotion, therefore improving baseplate fixation [69].

An FE study of RTSA showed that inferior tilt of the glenoid baseplate increased both micromotion and bone stress, which adversely affected implant fixation and longevity (Figure 5) [44]. While this study employed homogeneous isotropic bone material properties, more recent models have derived subject-specific inhomogeneous bone properties. For example, a FE study derived bone density and elastic modulus data from Hounsfield unit values obtained from computed tomography (CT) scans [70]. It showed that joint lateralisation significantly increased shoulder adduction range of motion, but this led to an increase in bone-implant micromotion. Further, reducing the number of fixation screws increased the negative effect of lateralisation on implant fixation.

When implementing the strain-induced bone remodeling algorithm proposed by Weinans et al. [71], simulated strains from anatomical FE models based on patient CT scans can be employed to investigate changes in bone density. A study showed that inferior positioning of the glenoid component did not increase the bone-implant micromotion, but inferior tilt of the glenoid component led to significantly higher bone-implant micromotion. Central positioning of the glenoid component was also found to cause a reduction in the apparent density of the bone in the lateral-inferior glenoid, while translating the glenoid component inferiorly led to bone loss in the lateral-middle region of the glenoid [20]. This validated model was later used to evaluated stress-shielding in RTSA and its influence in glenoid component loosening. Strain-induced bone resorption was observed, beginning at the peg of the implant and around the resected bone surface, which may ultimately lead to postoperative bone loss and glenoid loosening [72].

### 3.3. Prosthetic Joint Instability

Prosthetic instability leading to joint dislocation is one of the most frequent complications after RTSA, occuring in 4.7% to 31.0% of cases [17,63,73], and commonly leading to revision surgery [64]. Instability after RTSA has been related to inadequate soft tissue constraint, deltoid dysfunction, rotator cuff tears, incorrect implant sizing, implant malpositioning, and infection [17,74].

At present, there have been relatively few computational modeling studies exploring prosthetic joint instability after RTSA. An early FE analysis evaluated the effect of humeral component positioning on joint stability after RTSA using a constant joint compressive force with variable shear [41]. Joint stability was quantified as the ratio of the shear force needed to dislocate the joint to the applied compression force. It was found that 20° was the optimal degree of retroversion of the humeral stem to maintain implant stability. Another study employed a more detailed FE model to evaluate the influence of glenoid component version and humeral polyethylene liner rotation on subluxation and scapular notching [43]. Soft tissue tension around the glenohumeral joint capsule was modeled as spring elements placed symmetrically around the joint centre, and the deltoid was modeled as a cable and pulley system (Figure 6) [43,75]. It was found that neutral glenoid component version produced the least amount of subluxation, and posterior rotation of an asymmetric humeral polyethylene insert also reduced subluxation. This FE model was also used to assess the effects of variable rotator cuff deficiency on impingement after RTSA [76]. The results suggested that rotator cuff deficiency was associated with subluxation and impingement between scapula and humeral components. Specifically, posterior subluxation was associated with infraspinatus deficiency, while anterior subluxation was related to subscapularis deficiency. Another FE study used a similar modeling approach to evaluate the effects of several implant design and surgical factors on shoulder subluxation propensity following scapular impingement [77]. It was found that glenoid lateralisation and subscapularis repair reduced impingement-related subluxation.

To simulate physiological joint loading, one study built multi-body models to investigate the influence of rotator cuff tears on joint stability after RTSA [28] (Figure 1). A rigid body model was used to predict muscle forces during scapular-plane abduction and sagittal-plane flexion under different rotator cuff tear conditions, and these muscle forces were then used as inputs into an FE model of the reverse shoulder. It was shown that full-thickness tears to the infraspinatus significantly reduced glenohumeral joint compression by disrupting the rotator cuff muscle transverse-plane force couple, which may ultimately lead to glenohumeral joint instability.

### 3.4. Acromial and Scapular Spine Fractures

Acromial and scapular spine fractures are debilitating complications of RTSA reported in 0.8% to 10.2% of patients [78,79]. RTSA increases the deltoid moment arm and its nominal tension by medialising the centre of rotation and inferiorising the humerus. This may ultimately result in increased stresses and strains on the acromion and scapular spine, increasing the risk of fracture [6]. Other risk factors have been found to include a thin acromion, use of the onlay humeral stem, glenoid baseplate screw position/length, abnormal kinematics after RTSA, and osteoporosis [17,78,80,81]. Patients with acromial and scapular spine fractures may experience inferior clinical outcomes, including lower American Shoulder and Elbow Surgeons (ASES) scores, reduced ROM, and increased risk of revision arthroplasty [82].

To date, there have been relatively few studies investigating acromial fracture after RTSA using computational modeling and simulation. One study employed rigid body modeling and in vitro testing to evaluate change in length of the deltoid in ten cadaveric shoulders after RTSA. This study found that RTSA resulted in deltoid lengthening, which was exacerbated by humeral onlay lateralisation and glenosphere lateralisation. Lengthening of deltoid was shown to increase the strains on acromion and scapular spine and may lead to increased risks of acromial and scapular spine fractures, particularly when deltoid lengthening exceeds 25 mm [83]. One study used multi-body modeling to determine the effect of humeral and glenoid component positioning on acromial stresses [50]. Rigid body models of 10 shoulders after RTSA were used to simulate deltoid muscle forces during abduction motion, and the predicted muscle forces were subsequently used in FE analysis to calculate the acromial stress distribution. The results suggested that inferior and medial positioning of the glenosphere led to a decrease in acromial stress, while lateralisation of the glenosphere significantly increased the acromial stress and thus the potential risk of acromial fracture. These multi-body shoulder models were later used in a study to evaluate the effects of variations in arm loading and plane of elevation on acromial stresses [49]. It was found that abduction led to higher peak acromial stresses than scaption and forward elevation. Inferior positioning of the glenoid decreased the acromial stress in each scenario, while glenoid lateralisation consistently increased the acromial stress. The effect of variation in humeral offset was shown to be inconsequential with respect to changing the stresses at the acromion.

Another multi-body modeling study simulated deltoid and rotator cuff muscle loading to investigate the effect of acromion size on the risk of acromial fracture after RTSA (Figure 7) [51]. It was found that small acromion sizes were associated with higher stresses and therefore a higher risk of acromial fracture. Another study used a similar modeling technique to determine the influence of lateralisation and bone quality on acromion stresses [48]. The results showed that glenosphere lateralisation led to a higher risk of acromial and scapular spine fractures, which was exacerbated in patients with poor bone quality.

## 4. Experimental and Clinical Validation of Computational Reverse Shoulder Models

Computational models are validated to provide a ground-truth reference to model predictions. To achieve model validation, model outputs are directly compared against equivalent quantities that are measured in vivo, or using cadaveric or other testing techniques in the laboratory setting. Validation data commonly include measurements of joint reaction forces, muscle moment arms, muscle lines of action, and bone strain. Model validation is distinct from the process of model development, which may also employ experimental measurements of muscle architecture and bone and joint anatomy.

Computational models of the reverse shoulder that are used to predict shoulder ROM can be validated against experimental measurements of kinematics [84]; muscle moment arms compared against in vitro measurements, which are commonly obtained using the tendon excursion method [4,5]; muscle line of action data validated using experimental measurements [85]; muscle recruitment and timing of muscle activities validated against EMG recordings on patients [36]; and joint force predictions validated using cadaveric simulations [85].

Validation of FE models frequently involves direct comparison between simulated and measured joint contact areas and forces, joint moments, and bone/implant displacements and strains. For example, impingement site location and load magnitude has been validated against data obtained from cadaveric simulations of joint motion [86]; and simulated implant micromotion [68,87] and strains [72] validated using experimental simulations of bone and joint loading in the laboratory setting.

Validation of computational model outputs against clinical outcomes is critical for benchmarking model predictions, and numerous examples of this may be found in the extant literature. For example, findings of FE and multi-body modeling studies of the reverse shoulder have shown rotator cuff deficiency to be associated with joint instability [28,76], which is consistent with clinical outcomes of RTSA. Specifically, in patients with rotator cuff tear arthropathy treated with RTSA, dislocation is one of the most common complications and causes of revision surgery [11,88,89]. Rigid body model simulations have shown that inferior positioning and lateralisation of the glenosphere improves active shoulder ROM after RTSA and minimizes scapular notching [56,57,60], which is in agreement with RTSA patient cohort studies [90,91,92]. Furthermore, FE model simulations suggest that inferior positioning of the glenoid component does not lead to higher bone-implant micromotion [20], a finding supported by a clinical follow-up study of component loosening [90]. However, while computational modeling studies have revealed that glenosphere lateralisation is related to larger bone-implant micromotion and higher acromial stress [48,49,50,70], a clinical study suggests no significant differences in complication rates between medialized and lateralized glenosphere designs [91].

## 5. Limitations of Reverse Total Shoulder Arthroplasty Computational Modeling

While computational modeling may provide a basis for estimates of muscle and joint function, even the most sophisticated anatomical models are limited by the degree to which they replicate bone and joint biomechanics in vivo. Virtually all rigid body models neglect glenohumeral joint translation, and represent the glenohumeral joint as a three-degree-of-freedom constrained ball and socket or spherical joint [21,24,30,32,33,34,35,36,37,38,40,93]. This simplification limits the degree to which such a model can predict joint stability, since joint subluxation cannot be explicitly simulated. In addition, scapular notching is known to occur from repetitive contact between the humeral component and scapula during shoulder adduction; however, this phenomenon is frequently simplified in computational modeling through ‘contact’ detection algorithms, and progressive bone erosion is rarely modeled. Future studies ought to consider abutment-type and friction-type impingement, and the factors that lead to the evolution of notching-related bone erosion, including implant design, kinematics, and force. Another limitation of rigid body models is that muscles are represented as line segments, without volume or thickness. This means that the contact interaction between multiple muscles, as well as between muscle and bone/implant, and the muscle-tendon barrier effect, are neglected. While muscle and bone volumes can be reconstructed from medical imaging, such as CT and magnetic resonance imaging (MRI), this process is time consuming and represents a significant bottleneck in the model development pipeline. This has significantly limited integration of patient-specific muscle and bone geometry into rigid body models to date.

A further limitation of rigid body models is in representation of musculotendon parameters, including muscle force-length and force-velocity relations. This includes musculotendinous parameters optimum fibre length, tendon slack length, maximum fibre shortening velocity, and maximum isometric muscle force. These parameters are typically scaled from a generic model for practical reasons; however, it has been shown that subject-specific parameters, which can be calculated using data from a series of isokinetic and isometric contractions on a given subject, are markedly different from scaled parameters [31]. Furthermore, calculations of muscle and joint loading are sensitive to values of these parameters. Accurate estimation of muscle and joint forces in patients that have had RTSA may therefore be difficult or impossible to obtain without detailed experimental data derived from the patient and efficient computational tools required to estimate muscle force- and length-tension properties.

A perennial challenge in FE model simulations is in determining appropriate bone material properties and in deriving internal and external loads and boundary conditions. Some FE models represent bone as a rigid body or prescribe isotropic linear elastic material properties, which can simplify the modeling and simulation time when the focus of the analysis is on the stress and strain responses of the implant [41,61,62,75]. However, when the internal load response of bone is critical, for example when investigating bone remodeling or scapula-implant micromotion [20,70,72], realistic, physiologically relevant bone material properties are required. It has been shown that prescribing bone material properties using density data derived from CT produces joint displacements and reaction forces that agree well with experimental simulations of loading [94]. Simplified loads and boundary conditions are commonly used, facilitating model validation and providing reasonable estimates of bone-implant interactions [68,69,86,87]. However, when simulating the overall influences of RTSA on shoulder biomechanics, including changes in muscle and joint forces and the subsequent effects on bone-implant interactions, more physiological meaningful load boundary conditions are required, and these typically come from independent rigid-body model simulations [28].

## 6. Future Opportunities in Computational Modeling of Reverse Total Shoulder Arthroplasty

Advances in medical imaging, including bi-plane X-ray fluoroscopy and radiostereometric analysis, provide capability to accurately measure shoulder kinematics in vivo, including both glenohumeral joint rotations and translations [95,96]. These data can ultimately be used to drive patient-specific simulations of joint function during activities of daily living, which is critical for elucidating the nature of joint contact, such as the centre of joint contact pressure and peak contact stress. Measurements of implant micromotion may also be obtained and used for validating model predictions of implant stability and longevity, for example, as derived from radiostereometric analysis.

Evaluation of patient-specific anatomy for model development may require segmentation of medical images, which can be time-consuming and a significant barrier. Statistical shape modeling (SSM), which quantifies modes of geometric variation across a population through principal component analysis [97], has been employed in rapid reconstruction of 3D bone geometry without the need for segmentation [98,99]. SSM has the potential to be rapidly deployed for subject-specific modeling and surgical planning. Combining this technology with 3D printing may lead to faster development of personalized or ‘custom’ medical implants for the treatment of end-stage conditions of bone and joints where standard off-the-shelf implants are not suitable, for example large bone loss, tumour resection, congenital abnormalities, and trauma.

SSM has also been used to run virtual clinical trials of hip replacements across diverse cohorts [100], and investigate the effects of patient and surgical factors on the performance of implants across a wide range of possible load scenarios without actually using large patient cohorts. Finally, machine learning may also provide new opportunities for model development, and virtual clinical trials. Gaussian processes have already been used to rapidly predict postoperative micromotion given surgical approaches [101], suggesting that machine learning may open new possibilities for prediction of muscle and joint functions after RTSA.

In the future, computationally efficient personalized musculoskeletal modeling frameworks that combine both rigid-body and FE modeling into a single platform may provide a more efficient strategy for evaluating implant functional performance, and open the door to use in the clinical setting. Rigid-body musculoskeletal models that encapsulate joint translations and subject-specific muscle and joint properties may ultimately eliminate the need for computationally intensive FE modeling and provide greater insights into factors that drive glenohumeral joint stability after RSTA.

## 7. Conclusions

Computational models of the shoulder after RTSA are classified as either rigid body, FE, or multi-body models. They are developed to enable evaluation of quantities that would otherwise be difficult or impossible to measure, including implant stresses and strains, bone-implant micromotions, as well as muscle and joint forces. The effects of implant design, intraoperative component positioning, and surgical technique on postoperative shoulder biomechanics have been investigated using computational shoulder models, and the findings have been used to mitigate debilitating complications of RSTA that include scapular notching, component loosening, instability, and acromial fracture. The most significant challenge associated with the development of computational models is in the encapsulation of patient-specific anatomy and surgical planning. Advances in multibody modeling, SSM and artificial intelligence will facilitate a new generation of subject-specific models for implant design, surgical planning, and rehabilitation prescription, with future applications in both the research and clinical settings.

## Figures and Tables

**Figure 1 jcm-10-05336-f001:**
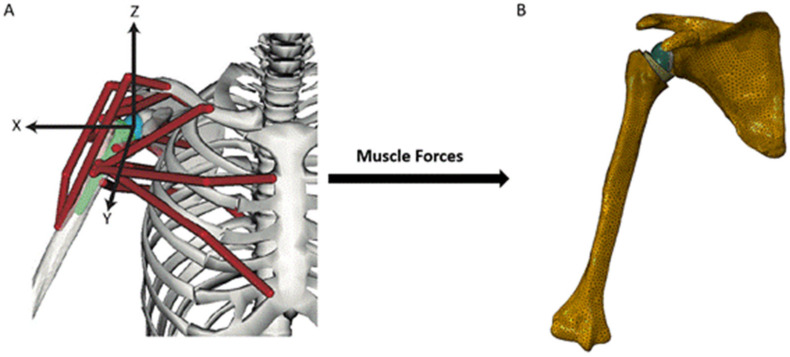
A multi-body modeling framework used to investigate joint stability after RTSA [28], including a rigid body model to simulate muscle and joint loading at the shoulder (**A**), and a FE model to calculate internal stresses and strains at the bone and implant (**B**). The muscle and joint forces calculated from the rigid body model were input into the finite element model and used to drive simulations of joint loading during activities of daily living.

**Figure 2 jcm-10-05336-f002:**
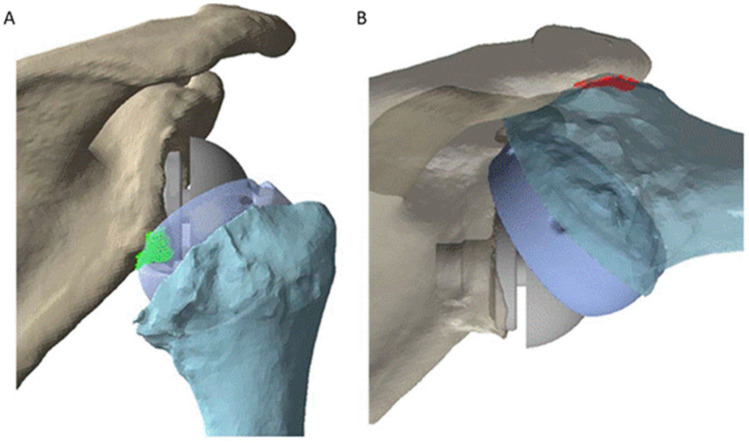
A rigid body model used to quantify postoperative shoulder ROM following RSTA [59]. A contact detection algorithm was employed to evaluate intra-articular (**A**) and extra-articular (**B**) impingement of the humeral component on the scapula.

**Figure 3 jcm-10-05336-f003:**
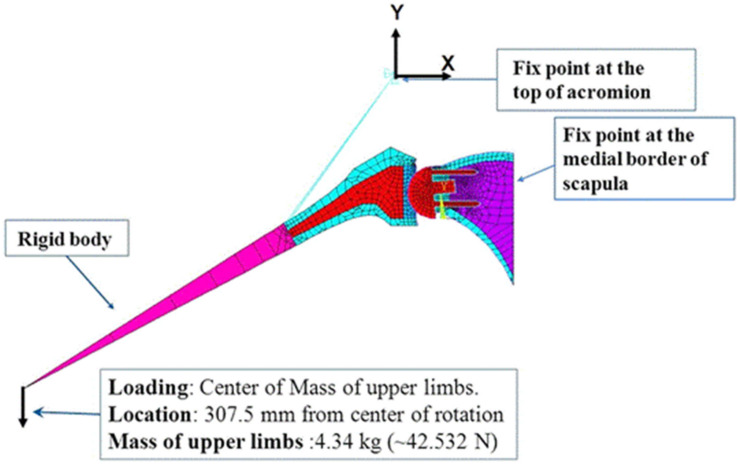
A 2D finite element model of the shoulder with a reverse prosthesis used to simulate internal stresses associated with the glenoid component during joint loading [61].

**Figure 4 jcm-10-05336-f004:**
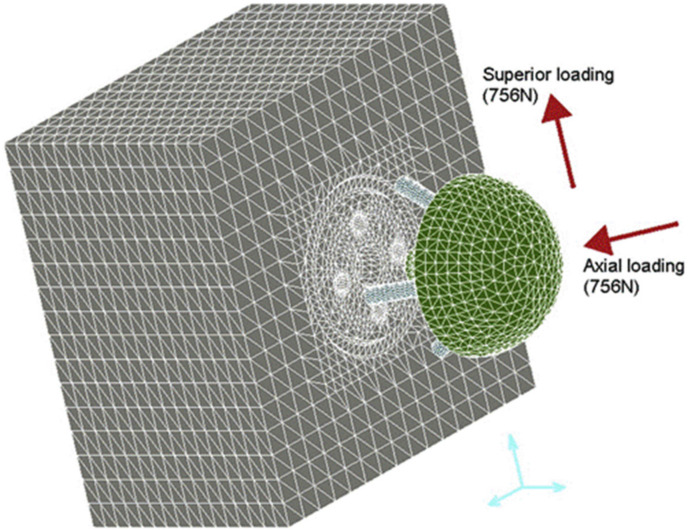
An FE model employed to quantify bone-implant micromotion during glenohumeral joint loading after RTSA [68]. The scapula bone was represented by a polyurethane block.

**Figure 5 jcm-10-05336-f005:**
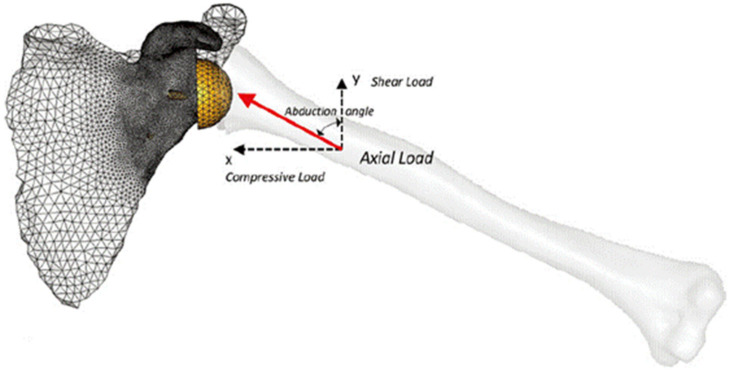
An FE model of the reverse shoulder developed using 3D anatomical bone geometry [44]. The bone geometry was digitally reconstructed from CT scans.

**Figure 6 jcm-10-05336-f006:**
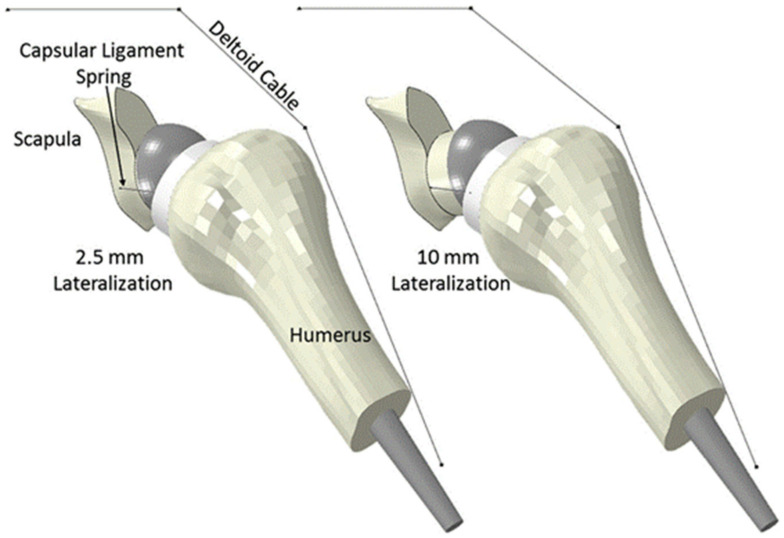
An FE model of the shoulder after RTSA that represented soft tissue tension around the glenohumeral joint capsule with spring elements, and the deltoid as a cable and pulley system [43,75].

**Figure 7 jcm-10-05336-f007:**
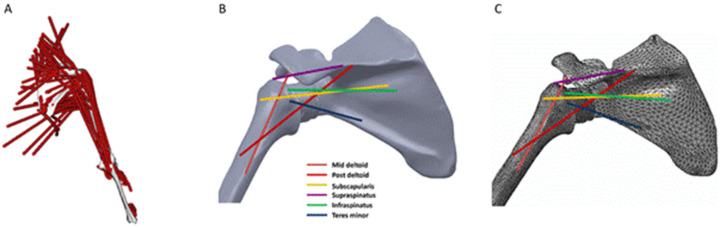
A multi-body model of the shoulder used to predict bone stress after RTSA [51]. Shown is the rigid body upper limb model (**A**), FE model with muscle lines of action (**B**), and the meshed FE model (**C**).

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
