# Peer review of "Complications of Reverse Total Shoulder Arthroplasty: A Computational Modelling Perspective"

_jcm, 2021, doi:10.3390/jcm10225336_

Round 1
Reviewer 1 Report
Meta-analysis of the literature regarding computational modeling studies on Reverse shoulder arthroplasty. Three biomechanical computer models (rigid body models, finite element models and multi body models) were used to investigate the 4 more common complications after RSA: scapula notching, glenoid loosening, dislocations and scapula spine fracture. Results are not new and have already been reported in the literature but this article is an interesting overview of what can be obtained from biomechanical computer models.
This literature provides a basis for future direction in computational modeling of the reverse shoulder.
Bibliography:
References 14 and 50 are the same.
Ref 45: What is “shoulder friends” ?
Reviewer 2 Report
Dear authors
With this manucrit, you review the literature about the contribution of computational modeling in the understanding and prevention of complications of RSA.
A better structuring of the manuscript is necessary to allow an easier reading more understandable for the reader.
As described below, some notions are missing. Likewise, some relevant studies could be cited.
It would also be interesting to cite clinical studies confirming or rejecting the conclusions of the theoretical studies cited in the manuscript.
My others comments :
48 - A recent study of RSA using an onlay-design 145° humeral stem identified an increased prevalence of scapular spine fracture compared with the standard Grammont design (~4–5% versus ~1%).
Ascione F, Kilian CM, Laughlin MS, Bugelli G, Domos P, Neyton L, et al. Increased scapular spine fractures after reverse shoulder arthroplasty with a humeral onlay short stem: an analysis of consecutive cases. J. Shoulder Elbow Surg. 2018 Dec;27(12):2183–2190. doi:10.1016/j.jse.2018.06.007
98 - In this paragraph about notching, authors discuss both ROM and notching. The limitation of ROMs and nocthing are not necessarily due to the same impingements. 2 paragraphs seem necessary or rename the paragraph with 2 differents parts : one about ROM and one about notching.
About the notching : clearly present the 3 factors that reduce the risk of notching :
1 – Glenoid inferiorization
2 – Glenoid lateralization
3 - Neck Shaft angle
It could be interesting to cite a few clinical studies confirming or denying the results of these computational studies.
99 - Notching represents a spectrum of impingement, which begins on the inferior part of the scapular pillar and spreads to the posterior part of the pillar.
Ladermann A, Gueorguiev B, Charbonnier C, et al. Scapular notching on kinematic simulated range of motion after reverse shoulder arthroplasty is not the result of impingement in adduction. Medicine (Baltimore). 2015;94(38):e1615.
102 – The impact of notch on RSA results and complications remains debated
Kohut G, Reuther F, Joudet T, Kääb MJ, Irlenbusch U. Inverted bearing reverse total shoulder arthroplasty: Scapular notching does not affect clinical outcomes and complications at mid to long-term. J Shoulder Elbow Surg. 2021 Oct 14:S1058-2746(21)00726-6. doi: 10.1016/j.jse.2021.09.010. Epub ahead of print. PMID: 34656780.
107 – As line 99 : notching should not be limited to a simple impingement between the humeral tray and the scapular pillar during adduction. Notching represents a spectrum of impingement, which begins on the inferior part of the scapular pillar and spreads to the posterior part of the pillar during the extension. Not only an abutment during the adduction. It’s a limit of several biomecanic or computing studies.
134 – A relevant study about notching
Werner BS, Chaoui J, Walch G. The influence of humeral neck shaft angle and glenoid lateralization on range of motion in reverse shoulder arthroplasty. J Shoulder Elbow Surg. 2017 Oct;26(10):1726-1731. doi: 10.1016/j.jse.2017.03.032. Epub 2017 May 17. PMID: 28528016.
143 – 155 : This is not the subject of this paragraph but the subject of the next one.
159 – The role of the inferiorization of the glenoid component in glenoid loosening is not discussed. It would be interesting to mention this and note the contradictory results between theoretical studies and clinical studies.
246 – Don’t forget spine fractures
Shah SS, Gentile J, Chen X, Kontaxis A, Dines DM, Warren RF, Taylor SA, Jahandar A, Gulotta LV. Influence of implant design and parasagittal acromial morphology on acromial and scapular spine strain after reverse total shoulder arthroplasty: a cadaveric and computer-based biomechanical analysis. J Shoulder Elbow Surg. 2020 Nov;29(11):2395-2405. doi: 10.1016/j.jse.2020.04.004. Epub 2020 Jun 9. PMID: 32669200.
315 - Remind readers that RSA lengthens the deltoid which can interfere with its function. This parameter is difficult to model
